



# Introducing the Extended Volatility Range Proton-Transfer-Reaction Mass Spectrometer (EVR PTR-MS)

Felix Piel[1,2,+], Markus Müller[1], Klaus Winkler[1], Jenny Skytte af Sätra[3,*] and Armin Wisthaler[2,3]

[1] IONICON Analytik, Innsbruck, Austria
     [2] Institute for Ion Physics and Applied Physics, University of Innsbruck, Innsbruck, Austria
     [3] Department of Chemistry, University of Oslo, Oslo, Norway
     [+] Now at: Department of Chemistry, University of Oslo, Oslo, Norway
     [*] Now at: Norwegian Environment Agency, Oslo, Norway

*Correspondence to*: Armin Wisthaler (armin.wisthaler@kjemi.uio.no)

**Abstract**. Proton-transfer-reaction mass spectrometry (PTR-MS) is widely used in atmospheric sciences for measuring volatile organic compounds in real time. In the most widely used type of PTR-MS instruments, air is directly introduced into a chemical

ionization reactor via an inlet capillary system. The reactor has a volumetric exchange time of ~0.1 s enabling PTR-MS analyzers to measure at a frequency of 10 Hz. The time response does, however, deteriorate if low-volatility analytes interact with surfaces in the inlet or in the instrument. Herein, we present the "Extended Volatility Range" (EVR) PTR-MS instrument which mitigates this issue. In the EVR configuration, inlet capillaries are made of passivated stainless steel and all wetted metal parts in the chemical ionization reactor are surface-passivated with a functionalized hydrogenated amorphous silicon

coating. Heating the entire set-up to 120°C further improves the time-response performance.

We carried out time-response performance tests on a set of 29 analytes having saturation mass concentrations $C^0$ in the range between $10^{-3}$ and $10^5\,\mu g\ m^{-3}$. 1/e-signal decay times after instant removal of the analyte from the sampling flow were between 0.2 and 90 s for gaseous analytes. We also tested the EVR PTR-MS instrument in combination with the CHARON particle inlet, and 1/e-signal decay times were in the range between 5 and 35 s for particulate analytes. We show on a set of exemplary

compounds that the time-response performance of the EVR PTR-MS instrument is comparable to that of fastest flow tube chemical ionization mass spectrometers that are currently in use. The fast time response can be used for rapid (~ 1 min equilibration time) switching between gas and particle measurements. The CHARON EVR PTR-MS instrument can thus be used for real-time monitoring of both gaseous and particulate organics in the atmosphere. Finally, we show that the CHARON EVR PTR-MS instrument is capable of detecting highly oxygenated species (with up to eight oxygen atoms) in particles

formed by limonene ozonolysis.



## 1 Introduction

The Earth's atmosphere contains a plethora of organic compounds, both in the gas and in the particulate phase (Goldstein and Galbally, 2007). Atmospheric organic compounds vary widely in their physico-chemical properties (*e.g.*, volatility, polarity, solubility), which makes their comprehensive measurement challenging (Heald and Kroll, 2020).

Proton-transfer-reaction mass spectrometry (PTR-MS) is widely used in atmospheric sciences for measuring volatile organic compounds (Hansel et al., 1995; de Gouw and Warneke, 2007; Yuan et al., 2017). In PTR-MS, air is directly introduced into an ion-molecule reactor wherein organic molecules ionize in collisions with hydronium ($H_3O^+$) ions. An electric field is applied across the ion-molecule-reactor, which is thus commonly referred to as the drift tube. Reagent and analyte ions are extracted from the drift tube and analyzed in a mass spectrometer.

One of the main advantages of PTR-MS is its rapidness. The drift tube has a volumetric exchange time of ~0.1 s enabling PTR-MS analyzers to measure at a frequency of 10 Hz (Müller et al., 2010). The time response does, however, deteriorate if low-volatility analytes interact with surfaces in the inlet or in the instrument. PTR-MS users have mitigated this problem by i) operating the drift tube at elevated temperature (Mikoviny et al., 2010), ii) increasing the flow through the drift tube (Breitenlechner et al., 2017; Krechmer et al., 2018) and iii) minimizing the wall collisions of analyte molecules (Breitenlechner

et al., 2017). The problem of analyte adsorption becomes more pronounced when particles are analyzed with the *Chemical Analysis of Aerosols Online* (CHARON) inlet (Eichler et al., 2015; Müller et al., 2017). After particle vaporization, low-volatility gases adsorb onto surfaces in the vaporizer, in the transfer line from the vaporizer to the drift tube and in the drift tube itself. This slows down the instrumental response significantly (Piel et al., 2019).

A second major advantage of PTR-MS is that $H_3O^+$ ions protonate both non-oxidized and oxidized organic analytes with
similar and high efficiency. $H_3O^+$ ion chemistry thus detects a wider spectrum of analytes than any other chemical ionization method for atmospheric organic carbon. It must, however, be ensured that analyte molecules do reach the ionization region and are not lost in the inlet line (*e.g.* Pagonis et al., 2017; Deming et al., 2019). If a long, low-flow and unheated polytetrafluoroethylene (PTFE) inlet line is used, the erroneous conclusion may be drawn that PTR-MS analyzers only detect singly and doubly oxygenated organic species (Riva et al., 2019).

Herein we will demonstrate how the use of inlet capillaries made of passivated stainless steel (SS) and the passivation of all wetted metal parts in the drift tube improves the performance of PTR-MS analyzers. We have observed a significantly improved measurement performance for low-volatility analytes. The instrument set-up described herein has thus been named "*Extended Volatility Range*" (EVR) configuration.

## 2 Experimental

### 2.1 The EVR PTR-MS instrument

The PTR-MS instrument has been described in detail elsewhere (Yuan et al., 2017; and references therein). The data presented herein were obtained with two state-of-the-art CHARON PTR-MS analyzers (models PTR-TOF 4000X2 and PTR-TOF 6000X2) produced by Ionicon Analytik (Innsbruck, Austria). In their conventional set-up, these analyzers include inlet capillaries made of polyether ether ketone (PEEK) and a drift tube plus ion funnel consisting of electropolished SS drift rings
and PTFE spacers. PEEK and especially SS are known to adsorb certain analytes. In an effort to optimize the instrumental time response, we have replaced all PEEK capillaries by treated SS capillaries. The SS capillaries were subject to a metal surface passivation process in which a functionalized hydrogenated amorphous silicon coating is applied for minimizing analyte-surface interactions. The same surface passivation was also applied to all wetted SS parts in the drift tube. In the CHARON inlet, the vaporizer and transfer tube from the vaporizer to the drift tube were surface-passivated. The inlet capillary
and drift tube temperature ($T_{drift}$) can be varied from room temperature to 120°C.





### 2.2 Performance assessment of the EVR PTR-MS instrument

A laboratory study was carried out to measure signal decay times in the EVR PTR-MS analyzer for 29 analytes listed in the Supplement (Tab. S1). A single analyte was supplied in steady concentration to the analyzer and instantly switched off. $\tau_{1/e}$ was measured as the time that evolved until the analyte signal decayed to 36% of the initial stable signal intensity. $\tau_{90}$ was

measured as the time that evolved until the signal decayed to 10% of the initial value. A stable gaseous analyte concentration, which is herein denoted with the subscript (g), was generated by placing a spatula tip of the solid sample into a 100 ml glass vial. The vial was heated and flushed with zero air (RH ~ 30%). Heating temperatures ranged from 50 to 120°C, depending upon the melting point of the analyte. The dynamic headspace of the vial was sampled through the gas inlet of the PTR-MS analyzer. Instrument and inlet were zeroed by overflowing the inlet with zero air (Fig. S1). A stable particulate analyte

concentration, which is herein denoted with the subscript (p), was generated by dissolving an aliquot of the solid sample in HPLC-grade water (Sigma-Aldrich Chemie GmbH, Taufkirchen, Germany). The solution was then nebulized with a home-built nebulizer. The nebulizer outflow was dried with two home-built diffusion dryers and gases were removed with an activated charcoal denuder (NovaCarb F, Mast Carbon International Ltd., Guilford, UK). The CHARON inlet and PTR-MS instrument were zeroed by diverting the sample flow through a high efficiency particulate air (HEPA) filter (Fig. S1).

Exemplary data from a field study were taken to show the signal response of the EVR PTR-MS analyzer when switching between the CHARON particle inlet and the gas inlet. The data were collected during a measurement campaign at the TROPOS Research Station Melpitz (Spindler et al., 2013) in Germany in February 2019.

A laboratory study was carried out for investigating the capability of the CHARON EVR PTR-MS analyzer to detect highly oxidized organic molecules in particles. For this purpose, we reacted ozone and limonene in a flow reactor to form secondary

organic aerosol (SOA). The reactor outflow was passed through an activated charcoal denuder (NovaCarb F, Mast Carbon International Ltd., Guilford, UK) for removing gaseous organics and subsequently injected into a 210 l steel barrel (Wilai GmbH, Wiedemar, Germany). The CHARON EVR PTR-MS analyzer sampled from this reservoir.

### 3 Results and Discussion

### 3.1 Signal decay in the EVR PTR-MS instrument

Fig. 1 shows the signal decay in an EVR PTR-MS analyzer ($T_{drift}$ = 120°C) after a steady supply of analyte was instantly switched off at t = 0 s. The upper panel shows exemplary data obtained for three gaseous analytes that were measured in separate experiments. The initial steady-state mixing ratios are reported in the figure legend. 4-nitrocatechol$_{(g)}$ (in dark yellow) exhibited a rather slow decay ($\tau_{1/e}$ = 27 s), *cis*-pinonic acid$_{(g)}$ (in red) decayed in a few seconds ($\tau_{1/e}$ = 2.4 s), and 2-tridecanone (in blue) dropped almost instantly to zero ($\tau_{1/e}$ = 0.3 s). For the latter, $\tau_{1/e}$ was close to the volumetric exchange time of the drift

tube (~ 0.1 s). The three exemplary compounds shown here cover the full three orders of magnitude span in $\tau_{1/e}$ ($10^{-1}$ to $10^{2}$ s) that was observed for gaseous analytes. For particulate analytes, $\tau_{1/e}$ ranged from a few seconds to a few tens of seconds. The lower panel of Fig. 1 shows exemplary data obtained for 2,7-dihydroxynaphthalene$_{(p)}$ (in dark yellow; $\tau_{1/e}$ = 15 s), levoglucosan$_{(p)}$ (in red; $\tau_{1/e}$ = 8.1 s) and nitrate$_{(p)}$ (in blue; $\tau_{1/e}$ = 4.4 s). The nitrate$_{(p)}$ signal originated from ammonium nitrate particles.


Fig. 2 summarizes the $\tau_{1/e}$ values measured for 21 gaseous analytes (upper panel) and 15 particulate analytes (lower panel). The color-coding and sizing are explained in the figure legend. Mixing ratios were quantified according to the procedure outlined in the Supplement of Müller et al. (2017) and were typically in the 0.1–10 ppbv range. Higher levels (up to 100 ppbv) were only used for a few compounds (vanillin$_{(g)}$, 2-tridecanone$_{(g)}$, 2,6-dimethoxyphenol$_{(g)}$, 4-nitroguaiacol$_{(g)}$, ammonia$_{(g)}$). We

typically measured $\tau_{1/e}$ at three different mixing ratios for each compound. Since these were in a rather narrow range, only


small changes in the instrumental time response were observed. We will thus not discuss any concentration dependence of $\tau_{1/e}$ here. We observed an increase in $\tau_{1/e}$ with decreasing saturation mass concentrations (log $C^0$) as a general trend, although with significant deviations for some compounds (Fig. S2). Glucose$_{(g)}$, for example, exhibited a much faster response than the sugar alcohols (xylitol$_{(g)}$, arabitol$_{(g)}$) despite having a similar log $C^0$. Structural effects may play a role here, since glucose is a cyclic

molecule, while the sugar alcohols are both linear. It is also important to note that the SIMPOL.1 method has not been validated for saccharides and that the calculated log $C^0$ may be inaccurate. 4-nitrocatechol, with a relatively high log $C^0$ of 4.2, was among the slowest responding gaseous analytes. This observation remains unexplained. With the CHARON inlet connected, $\tau_{1/e}$ was in the 5 to 20 s range for most analytes (Fig. 2, lower panel). Exceptions were diglycolic acid$_{(p)}$ and tartaric acid$_{(p)}$ with $\tau_{1/e}$ up to ~35 s. Notably, no obvious dependence of $\tau_{1/e}$ on log $C^0$ was observed. Levoglucosan, 2,7-dihydroxynaphthalene,

stearic acid, azelaic acid, diglycolic acid and vanillic acid were studied in both phases. The instrumental response was typically ~5 s slower in the particle measurements. The CHARON inlet has a larger surface area and a lower sample flow than the gas inlet. Stearic acid and azelaic acid responded faster with the CHARON inlet, which remains unexplained. The reader is cautioned that the $\tau_{1/e}$ values presented in Fig. 2 should not be taken as absolute and generally applicable values. The reported numbers should be seen as indicative estimates for the time response of state-of-the-art IONICON EVR PTR-MS instruments.

It is well known that analytes compete for surface adsorption with other matrix constituents such as water or other surface-affine compounds. All of our experiments were carried out with a single compound at one humidity level. We consider it beyond the scope of this work to investigate a matrix dependence of $\tau_{1/e}$. In previous work, we anecdotally observed that basic analytes exhibited a significantly slower time response when acidic samples had been sampled before. The sampling history was not considered in our study.

### 3.2 Drift tube temperature effects

We investigated the effect of $T_{drift}$ on $\tau_{1/e}$ for two gaseous analytes, 2-tridecanone$_{(g)}$ and *cis*-pinonic acid$_{(g)}$ (Fig. 3). In the case of 2-tridecanone$_{(g)}$ (upper panel), an increase in $T_{drift}$ from room temperature to 60 °C decreased $\tau_{1/e}$ from 24 s to 2 s. At $T_{drift}$ = 100°C, $\tau_{1/e}$ was 0.3 s approaching the volumetric exchange time of the drift tube. In the case of *cis*-pinonic acid$_{(g)}$ (lower panel), we only investigated the 80 to 120°C temperature range in which $\tau_{1/e}$ dropped from 6.5 to 2.4 s. For the two compounds

investigated, the decrease in $\tau_{1/e}$ can be explained by the increase in log $C^0$ with temperature. According to Epstein et al. (2010), a 15°C temperature rise increases log $C^0$ by ~1 in the 0 to 50°C temperature range. The effect becomes less pronounced at higher temperatures. For *cis*-pinonic acid, the SIMPOL.1 method yields a log $C^0$ increase from 5.2 to 6.2 in the 80 to 120°C temperature range. At 45°C, 2-tridecanone has roughly the same log $C^0$ as *cis*-pinonic acid at 120°C (Fig. S3). Consistently, we observed a similar time response for the two compounds at $T_{drift}$ = 45°C and $T_{drift}$ = 120°C, respectively. We conclude that

increasing $T_{drift}$ to 120°C is an effective way for reducing $\tau_{1/e}$ in EVR PTR-MS analyzers. Importantly, none of the compounds studied herein thermally decomposed at this temperature. This needs to be carefully assessed when a higher $T_{drift}$ is used or when thermally labile compounds are to be analyzed.

### 3.3 Response times of different online CIMS instruments

Fig. 4 compares the signal decay times (here reported as $\tau_{90}$ and not as $\tau_{1/e}$) of different online CIMS instruments to a set of

ketones, carboxylic acids and hydroxycarbonyls in the gas phase. The plot includes data from this study, which were obtained with two state-of-the-art IONICON PTR-MS analyzers. The data points in yellow were obtained with a conventional instrument, while the data points in dark red were collected with an EVR-type analyzer. Both instruments were operated at $T_{drift}$ = 120°C. The figure also includes literature data obtained with an old quadrupole PTR-MS instrument (qPTR-MS; in blue: Pagonis et al., 2017), with a PTR-ToF-MS analyzer from a different manufacturer (VOCUS$^{TM}$; in orange: Krechmer et

al., 2018) and with two iodide (I$^-$) CIMS instruments (in green: Liu et al., 2019; in light blue: Palm et al., 2019), respectively.





$\tau_{90}$ is plotted against log $C^0$ of the respective analyte (specified on the upper horizontal axis) which was calculated using the SIMPOL.1 method (Pankow and Asher, 2008). At $T_{drift}$ = 120°C, even the conventional PTR-MS analyzer responded almost instantly to $C_7$-, $C_{10}$- and $C_{13}$-ketones. For the $C_{13}$-species, the EVR-type instrument responded only ~20% faster than the conventional analyzer. The VOCUS™ instrument and the qPTR-MS analyzer exhibited a much slower response to these

ketones. We explain this by the fact that both of these instruments were operated at room temperature.

The superior performance of the EVR-type instrument becomes evident when measuring carboxylic acids. In the case of *cis*-pinonic acid$_{(g)}$, $\tau_{90}$ decreased by a factor of 30 when using an EVR PTR-MS analyzer instead of a conventional instrument. Stearic, azelaic and diglycolic acid, for which log $C^0$ ranges roughly from 1 to 2, exhibited response times in the range of 1 to 2 minutes. This is as fast as an optimized I⁻ CIMS instrument with reduced instrument wall interactions responded to

dihydroxycarbonyls with a log $C^0$~2 (Palm et al., 2019). A significantly slower response was recently reported for another I⁻ CIMS instrument (Liu et al., 2019).

### 3.4 Rapid switching between the gas inlet and the CHARON particle inlet

For certain applications, it is desirable to periodically switch between the gas and particle measurements. Fig. 5 shows exemplary data collected by a CHARON EVR PTR-MS instrument during ambient air measurements at a rural background

station in Germany.

The reader should focus on the transition from background to ambient particle measurements at 06:15:40 and the transition from particle to gas measurements at 06:21:40, which is when the slow signal response becomes most evident. The upper panel shows the time evolution of the $NH_4^+$ (*m/z* 18.034) and $NO_2^+$ (*m/z* 45.996) signals. In the CHARON inlet, ammonium nitrate particles evaporate to yield gaseous ammonia ($NH_3$) and nitric acid ($HNO_3$). Ammonia is detected in its protonated form, while

protonated nitric acid dehydrates upon protonation to yield the nitronium ion ($NO_2^+$). Both ammonia and nitric acid are particularly prone to adsorptive losses on stainless steel (Neuman et al., 1999; Nowak et al., 2007). In the EVR-type instrument, wetted surfaces do not include any untreated stainless steel, which results in a fast instrumental response to both compounds. Both the $NH_4^+$ and the $NO_2^+$ signal equilibrated within 1 minute when switching from HEPA to ambient CHARON measurements and from particle to gas measurements, respectively. The lower panel shows the evolution of the $C_4H_5O^+$ (m/z

69.033) and $C_6H_9O_4^+$ (m/z 145.049) signals within one measurement cycle. $C_6H_9O_4^+$ is the main ionic fragment from levoglucosan (Leglise et al., 2019). $C_4H_5O^+$ is believed to be a fragment of larger furanoid compounds in particles and protonated furan in the gas measurement. The instrumental response to these analytes was somewhat slower than to ammonium nitrate but equilibration still occurred within 1 minute. The instrumental response to levoglucosan is similar to what we observed in single compound measurements in the laboratory, suggesting that the presence of a complex matrix does not

negatively affect instrumental response times.

### 3.5 Detection of highly oxidized organic compounds

In an effort to test the capability of detecting highly oxidized species, we measured SOA generated from limonene ozonolysis with optimized instrumental settings (EVR, $T_{drift}$ = 120°C). In addition, we operated the instrument at low reduced electric field strength (E/N = 30 Td; 1 Td = $10^{-17}$ Vcm$^{-2}$) and with $NH_4^+$ as the reagent ion (Müller et al., 2020). With these instrumental

settings, ionic fragmentation is largely suppressed and highly oxidized organic molecules are detected in their ammonium adduct form (Müller et al., in preparation).

Figure 6 shows the normalized mass distribution of oxidized species detected in limonene/$O_3$ SOA as a function of #O, which ranged from 1 to 8. The majority of the highly oxidized compounds (#O ≥ 3) are monomers (#C ≤ 10, ~55% of the total mass). The most abundant ions observed fall into this category and can be assigned to limonene ozonolysis products previously



reported in the literature (Tab. S4). For the condensation products or dimers (#C > 10), ~7% of the total mass was composed of highly oxidized compounds. Compounds with #O > 6 were equally distributed between monomers and dimers. We detected compounds up to #O = 8, which accounted for ~ 0.5 % of the total measured mass concentration.

It is worth pointing out that some of the ions detected in limonene/$O_3$ SOA (Fig. S4) have been previously associated with analytes containing a hydroperoxy functional group (Hammes et al., 2019). Rivera-Rios et al. (2014) observed that

hydroperoxides efficiently decompose on the metal parts (inlet, drift tube) of PTR-MS instruments. Preliminary laboratory tests with cumene hydroperoxide and dicumylperoxide (*i.e.*, the only peroxides that are commercially available) indicate that this process does not occur in EVR-type instruments. An additional benefit of the EVR configuration may thus be that the metal-catalyzed decomposition of labile compounds (*e.g.*, peroxides and hydroperoxides) is eliminated or suppressed. More work is needed to confirm this preliminary finding.

## 4 Conclusion

We have described and characterized the novel EVR PTR-MS instrument, which exhibits a significantly improved time-response performance as compared to conventional IONICON PTR-MS analyzers. The time response of this optimized instrument is comparable to that of fastest flow tube CIMS instruments that are currently in use. This allows to rapidly switch between gas and particle measurements, making the CHARON EVR PTR-MS instrument the only direct sample introduction

CIMS instrument that can monitor gaseous and particulate organics in the atmosphere in real time. Besides being faster, the EVR PTR-MS instrument also allows to target new analyte classes such as highly oxygenated organic molecules and hydroperoxides. We believe that the CHARON EVR PTR-MS instrument will be a valuable tool for overcoming current challenges in the measurement of atmospheric organic carbon (Heald and Kroll, 2020).

### Author contribution

KW developed the EVR system and performed initial tests. MM, FP and AW designed the experimental studies. MM, FP and JSS carried out the experiments. FP performed the data analysis, with support from JSS and MM. FP and MM drafted the manuscript. AW wrote the final manuscript. All authors commented and accepted the final version of the manuscript.

### Conflicts of interest

MM, FP and KW work for IONICON Analytik, which is commercializing (CHARON) PTR-MS instruments. MM and AW

profit from a license agreement (CHARON inlet) between the University of Innsbruck and IONICON Analytik.

### Acknowledgements

We like to thank Laurent Poulain and Gerald Spindler from TROPOS for their support during the measurements in Melpitz. Measurements in Melpitz were supported by the European Union's Horizon 2020 research and innovation programme under grant agreement N°654109 (ACTRIS-2). FP has received funding from the European Union's Horizon 2020 research and

innovation programme under grant agreement N°674911 (IMPACT). Special thanks go to Magda Claeys for providing many of the substances studied in this work.

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

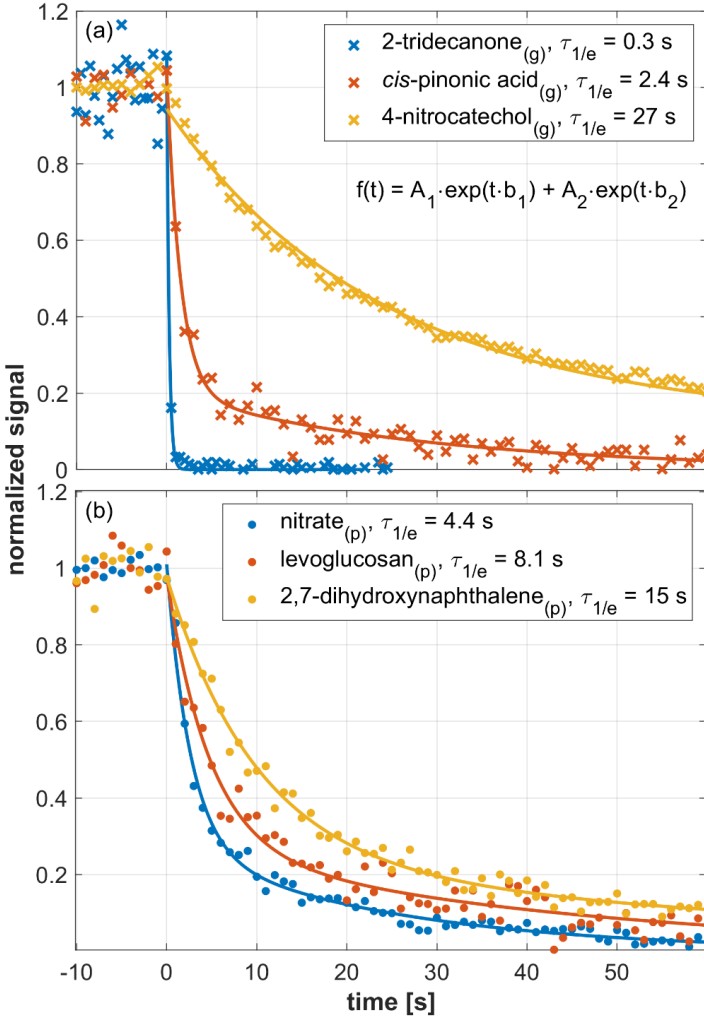

**Figure 1. Signal decay as observed in an in an EVR PTR-MS analyzer (T$_{drift}$ = 120°C) after a steady supply of analyte was instantly switched off a t = 0 s. The decay of gaseous analytes is shown in the upper panel, while the lower panel refers to particulate analytes. Initial steady-state mixing ratios were as follows: 2-tridecanone$_{(g)}$: 1 ppbv, *cis*-pinonic acid$_{(g)}$: 2.2 ppbv, 4-nitrocatechol$_{(g)}$: 30 ppbv, nitrate: 3 ppbv; levoglucosan$_{(p)}$: 1.2 ppbv, 2,7-dihydroxynaphtalene$_{(p)}$: 0.3 ppbv. Signals were fitted using a double exponential decay function (see insert in the upper panel). All fitting parameters are listed in Table S2.**


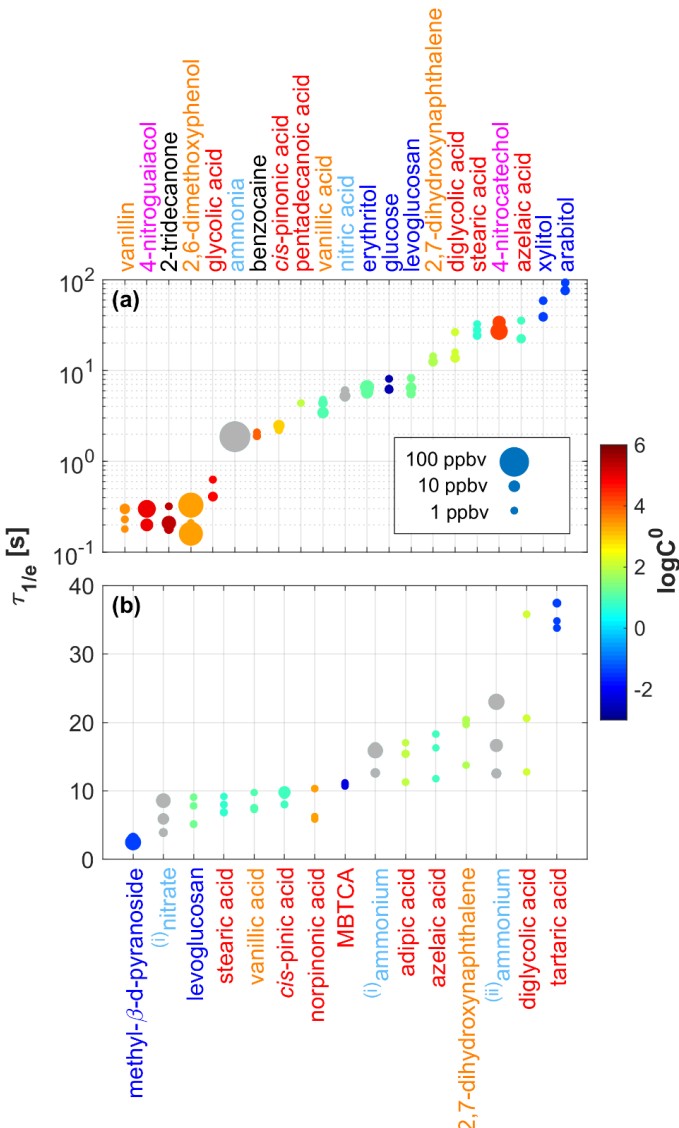

**Figure 2. Signal decay times (τ$_{1/e}$) measured for 21 gaseous analytes (upper panel) and 15 particulate analytes (lower panel). Analytes were grouped and color-coded into six classes: saccharides (in dark blue), carboxylic acids (in red), substituted phenols (in orange), nitroaromatics (in magenta), small polar molecules (in light blue) and others (in black). The size of the dots indicates the initial steady-state mixing ratio (0.1–100 ppbv) used in the respective experiment. The color code of the data points indicates the saturation mass concentrations (log C$^0$) of the analytes as calculated using the SIMPOL.1 method (Pankow and Asher, 2008).**

**(i) originating from ammonium nitrate, (ii) originating from ammonium sulfate**



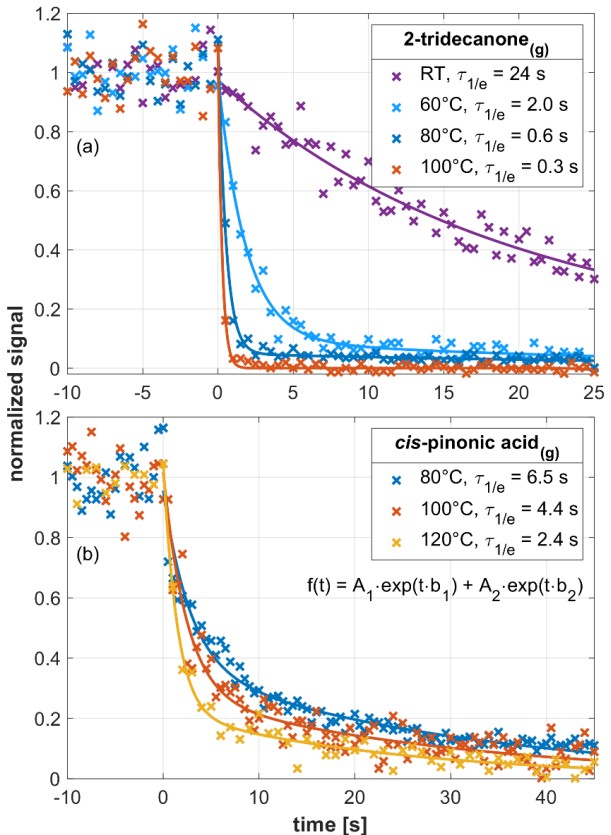

**Figure 3. Signal decay as observed in an EVR PTR-MS analyzer for 2-tridecanone(g) (upper panel)) and *cis*-pinonic acid(g), respectively, at different drift tube temperatures. Initial steady-state mixing ratios were as follows: 2-tridecanone(g): 1–1.3 ppbv; *cis*-pinonic acid: 2.2 ppbv. Signals were fitted using a double exponential decay function (see insert in the lower panel). All fitting parameters are listed in Table S3.**

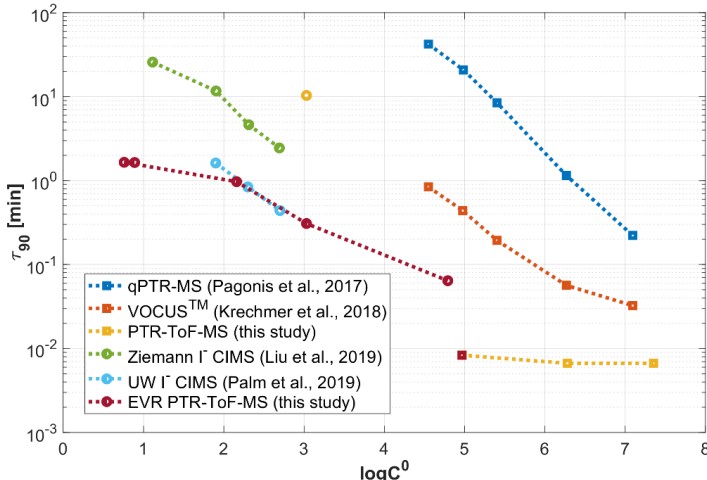

**Figure 4. Signal decay times (here reported as $\tau_{90}$) as observed in different online CIMS instruments for a set of ketones, carboxylic acids and hydroxycarbonyls. $\tau_{90}$ is plotted as a function of the SIMPOL.1-derived saturation mass concentration (logC⁰).**



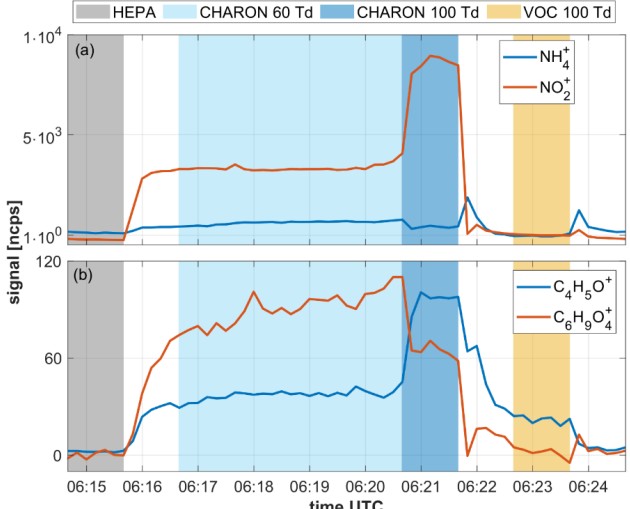

**Figure 5. Time series showing various analyte signals as recorded during a 10-minute measurement cycle of ambient air. The 10-min measurement cycle included: i) 1 min of instrumental background measurements with the CHARON inlet including a HEPA filter, ii) 4 mins of particulate measurements at an E/N of 60 Td, iii) 1 min of particulate measurements at an E/N of 100 Td and iv) 1 min of gas measurements at an E/N of 100 Td. The benefit of measuring particles at 60 and 100 Td is explained in Leglise et al. (2019) and is not discussed here. The CHARON inlet enriched the particle concentration by a factor of ~20, which explains the higher signal intensities in the particle measurement (in blue) as compared to the gas measurement (in yellow).**

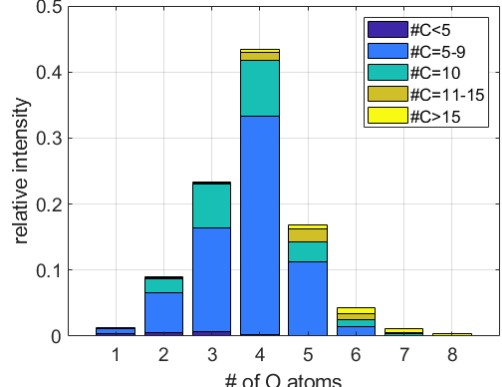

**Figure 6: Normalized mass distribution as a function of the number of oxygen atoms (#O) that was observed when the CHARON EVR PTR-MS analyzer sampled SOA generated from the reaction of limonene with ozone. The color code indicates the number of carbon atoms (#C). Mass concentrations were derived from the assumption that all analytes form ammonium adducts at the collisional rate (Zaytsev et al., 2019).**