# Peer review of "Introducing the Extended Volatility Range Proton-Transfer-Reaction Mass Spectrometer (EVR PTR-MS)"

_Atmospheric Measurement Techniques, 2020_

## Referee Comment (RC1) · Anonymous Referee #1 · 9 Sep 2020

Piel et al present the characterization of a new type of inlet for Ionicon PTR-MS instruments, which enhances the sampling of molecules less volatile than typically reported by such instruments. This is a good addition, since PTR has more potential than has been realized in the most commonly deployed instruments, where VOC detection has been the main focus. The paper is for the most part clear and well written, and the topic suitable for AMT. However, I have a few questions and comments to the authors, relating to both clarity and novelty, that need to be addressed before possible publication in AMT.

Major comments:

In the abstract, lines 18-20, the EVR configuration is presented primarily as an improvement by passivating surfaces, with "further improvement" by heating to 120 C.

However, as far as I can tell, there is no data presented in the manuscript about how much the passivated surfaces changed the response times. If the authors want to highlight the passivation, this type of information needs to be included in some form. Otherwise, the temperature increase, for which there is clear data (e.g. Fig. 3a), should be presented as the main improvement. And in that case, the novelty of this manuscript (and the EVR inlet) is questionable, as there are a wealth of PTR studies where inlets and inlet lines have been heated in order to improve response times and detection of less volatile species (see e.g. section 2.4.5 in Yuan et al., 2017, Chem. Rev.). The authors need to make very clear what exactly causes the improvement of response times in this manuscript, and how this is different enough from earlier work that it deserves publication in AMT.

Lines 55-58 at the end of the introduction do not even mention the temperature issue, suggesting that the material changes are the main topic of this manuscript. This requires verification.

A related point is the lack of any schematic diagram of the EVR in the manuscript. It may be understandable if the design itself did not change, but rather only materials were exchanged, yet it would still be beneficial for a reader to see a figure showing these changes. As it is, the only reference on PTR in section 2.1 is the Yuan et al (2017) review, which itself doesn't have a schematic of the exact system used in this work. This makes it very laborious for a reader to understand the changes, and consequently to properly assess the novelty of the changes and the manuscript itself.

Specific comments:

1. Lines 28-29: Since these measurements were done using NH4+ adducts, I don't think it should any longer be called "PTR-MS".

2. Lines 69-70. Since T_drift is used later as a parameter, it is important for a reader to have a clear picture of how the drift tube looks. Also here a schematic would be useful.

3. Lines 50-51: "H3O+ ion chemistry thus detects a wider spectrum of analytes than any other chemical ionization method for atmospheric organic carbon." This is a strong statement and would need a citation. Instruments like the NH4+-CI3-TOF or C3H7NH3+-APi-TOF (Berndt et al., 2018, Angew. Chem. Int. Ed. 2018, 57, 3820 –3824) seem to detect almost all organic compounds (including radicals) except hydrocarbons. Can the authors show references where H3O+ ion chemistry would have detected a broader spectrum than that?

4. Lines 73-75. These two sentences need to be reformulated. "was measured as the time that evolved until" is hard to understand.

5. Fig. S1. The part inside the red dashed box is presumably the inlet and drift tube of the PTR? This needs to be clarified, as it is not easy to read for someone not highly experienced with the system.

6. Lines 108-109: This leaves the reader with the question "why?".

7. Lines 133-134: This also leaves the reader with the question "why?".

8. Section 3.2: The discussion is about T_drift, and the text suggests that it is only the drift tube temperature that is changed. But Fig. S1 suggests that the entire inlet is one temperature-controlled entity. Please clarify. This again would be easier to understand if there was a proper schematic included.

9. Lines 138-139: This seems consistent, but Fig. S2 shows that the dependence of tau on C0 is very weak, with compounds of the same C0 easily having an order of magnitude or larger differences in tau. Were these two compounds selected to be shown because they happened to match?

10. Line 144: Why are you now shifting from tau_1/e to tau_90?

11. Line 145: It is unclear which data from this study is used in Fig. 4. This should be made clear, so that a reader would be able to compare responses compound by compound. In addition, there is a nice monotonic trend of tau vs C0 in Fig. 4, which is

hard to understand given the huge spread of points in Fig. S2. One is left wondering how the authors selected the 5 data points shown in Fig. 4 for the EVR.

12. Line 153: C13 ketones are referenced, but I do not know where I should be looking to find this data.

13. Fig. 4: It took me a while to realize that the unit of tau changes between Fig. 2 and Fig. 4, from sec to min. Why not keep them the same? Now both the time unit and the decay reference (1/e vs 90) change, and it makes things much harder to follow. I suggest to make all of these the same, as it would make the reading much smoother and avoid confusion.

14. Fig. 4: If I understand the plot correctly, the non-EVR PTR-TOF from this study seems to work much better than the EVR. All measured response times are on the order of 0.01min, while in the EVR all points are at 0.1 min or higher. Does this mean that the EVR setup has actually made the response times worse compared to the original design (as long as the inlet is heated)?

15. Line 155: The fact that temperature explains the major part of the differences, and this fact is only mentioned in one sentence in the main text, makes Fig. 4 very misleading. For example, one would read from Fig. 4 that the reason the PTR-TOF in this study was 3-4 orders of magnitude better than the PTR-qMS from Pagonis et al is related to the quad vs tof, since that is the only clear difference. Not to mention the comparison to other instruments, which were not run at elevated temperatures. Please put the operating temperatures into the figure legend, since these values are the most critical parameter to understand the major differences in the figure.

16. Lines 171-172: Again, the fast changes are attributed to the materials, and temperature is not mentioned at all. This needs to be clearly validated before making this claim.

17. Conflict of interest statement: Ionicon analytik is said to be "commercializing

(CHARON) PTR-MS", but they are also advertising directly the EVR setup presented here (https://www.ionicon.com/accessories/details/extended-volatility-range-evr). Why is this not mentioned here?

Technical corrections:

1. Line 97 and line 107: Caption, not legend.

---

## Referee Comment (RC2) · Anonymous Referee #2 · 1 Oct 2020

This manuscript presents the development and characterization of a new type of inlet for PTR-MS instruments that allows for detection and quantification of less volatile compounds. The authors demonstrate how the new instrument can be integrated with the CHARON setup and used for online measurements of gas- and particle-phase organics. This kind of instrument development is quite valuable, and I think these data (including intercomparison of response times for various CIMS instruments) should be available for the CIMS community. However, on a whole the presented data and discussion are somewhat limited, and the paper does not cover some important parameters of the new instrument that are critical for its full characterization. While the paper is fairly well-written, it would require some major revisions before being published in Atmospheric Measurement Techniques.

**Major comments:**

1. The purpose and the focus of the manuscript are a little unclear. Is it just to estimate the signal-decay times for a new instrument or to assess its performance more holistically? If so, the effects of interactions with inlet walls and humidity should be discussed in this paper.

2. The effect of the drift tube temperature is interesting and important for more comprehensive evaluation of the instrument measurement capability, but is weakened by the very small number of compounds used to derive the conclusion that 120$^o$C is the optimal temperature at which the instrument should be operated. It would be of interest to conduct similar measurements with a larger set of compounds, especially with the ones that tend to thermally decompose at higher temperatures, such as hydroperoxides (i.e., cumene hydroperoxide and dicumylperoxide discussed later in the paper).

3. I do not fully understand how detection of particle-phase highly oxidized organic compounds produced via ozonolysis of limonene fits in this paper. The authors neither discuss the signal-decay times for these compounds nor try to estimate the respective wall losses. It has been shown several times that softer ionization techniques, such as $NH_4^+$ CIMS, can be used for detection of highly oxygenated compounds in the gas and particle phase (i.e., Hansel et al., 2018; Zaytsev et al., 2019). Hence, the authors should clarify why they present these data and how implementation of the new inlet improves detection and quantification of these compounds.

**Specific comments:**

1. Lines 49-54: From reading this paragraph one might get a false impression that PTR-MS has the best measurement capability among all CIMS instruments. While it is true that the ionization efficiency does not vary too much among oxidized and non-oxidized compounds, the overall measurement capability of PTR-MS instruments is significantly limited by ionic fragmentation and wall losses. Hence, the authors should edit this paragraph and make their description of PTR-MS more balanced.

2. Lines 55-58: The effects of drift tube temperature are not mentioned here, however they seem to play a fairly important role as outlined in the Discussion section. The particle-phase experiments should also be mentioned at the end of Introduction.

3. Section 2.1: This section is missing a schematic of the EVR PTR-MS instrument. I suggest moving Fig S1 to the main text and significantly expanding it to demonstrate what parts of the instrument were replaced or coated. As of now, these changes might not be obvious especially for a reader who is not fully familiar with IONICON PTR-MS instruments.

4. Line 95: Why did the authors use the double exponential decay for fitting the signals? How did the authors calculate $\tau_{1/e}$ from fitted parameters $b_1$ and $b_2$? I believe this is not explicitly discussed in the paper.

5. Figure 2: What do different circles/data points for the same compound represent? The authors should clarify this and discuss why the difference between some data points is fairly large, for example for 2,6-dimethoxyphenol and diglycolic acid it can be up to a factor of 2.

6. Lines 140-141: It would be beneficial if the authors could include high-resolution mass-spectra in the Supplement to demonstrate that studied compounds did not thermally decompose. Many of observed compounds are known to undergo ionic fragmentation (e.g., $C_6H_9O_4^+$ is an ionic fragment of levoglucosan as discussed later in the paper), so how do the authors know that there is no additional thermal decomposition resulting in formation of those fragments?

7. Section 3.3: The authors should provide a table in which they should list compounds that were used to compare performances of various instruments. What ketones, carboxylic acids and hydroxycarbonyls were used in this study?

8. Figure 4: It seems to me that the authors did not measure response times for the same compounds using a conventional PTR-MS and a new EVR PTR-MS as yellow and dark red points are located far from each other (for yellow dots 5<log $C^0$<7.3 while for dark red dots 0.5<log $C^0$<5). I suggest that the authors include additional data points on this figure to demonstrate how the performance of the EVR PTR-MS instrument compares with a conventional IONICON PTR-MS for the same group of compounds.

9. Figure 4: I agree with the Referee 1 that operational temperatures should be clearly stated in the legend of this figure.

10. Figure 6: The authors state that mass concentrations of observed compounds were calculated under the assumption that all of these compounds were detected at the collisional rate. The authors should clarify how this collisional rate was calculated. In addition, they should explicitly mention it in the text as this is a fairly important assumption and can strongly affect the authors' conclusion about mass yields of observed compounds.

11. Conflict of interest: I agree with the Referee 1 that the authors should mention the fact that IONICON has been advertising the EVR PTR-MS setup for quite some time now.

**Technical corrections:**

1. Line 307: remove "in an"

**References:**

1. Hansel, A., Scholz, W., Mentler, B., Fischer, L., Berndt, T. (2018). Detection of RO$_2$ radicals and other products from cyclohexene ozonolysis with NH$_4^+$ and acetate chemical ionization mass spectrometry. Atmospheric Environment, 186, 248-255.

2. Zaytsev, A., Breitenlechner, M., Koss, A. R., Lim, C. Y., Rowe, J. C., Kroll, J. H., Keutsch, F. N. (2019). Using collision-induced dissociation to constrain sensitivity of ammonia chemical ionization mass spectrometry (NH$_4^+$ CIMS) to oxygenated volatile organic compounds. Atmospheric Measurement Techniques, 12(3), 1861.

---

## Author Comment (AC1) · 15 Nov 2020

**Response to Reviewer #1:**

We thank the reviewer for carefully reading our manuscript and for providing highly valuable comments, which we have addressed in detail below.

**Major comments:**

*In the abstract, lines 18-20, the EVR configuration is presented primarily as an improvement by passivating surfaces, with "further improvement" by heating to 120 C. However, as far as I can tell, there is no data presented in the manuscript about how much the passivated surfaces changed the response times. If the authors want to highlight the passivation, this type of information needs to be*

[Figure]

*included in some form. Otherwise, the temperature increase, for which there is clear data (e.g. Fig. 3a), should be presented as the main improvement. And in that case, the novelty of this manuscript (and the EVR inlet) is questionable, as there are a wealth of PTR studies where inlets and inlet lines have been heated in order to improve response times and detection of less volatile species (see e.g. section 2.4.5 in Yuan et al., 2017, Chem. Rev.). The authors need to make very clear what exactly causes the improvement of response times in this manuscript, and how this is different enough from earlier work that it deserves publication in AMT.*

We have found data from two experiments wherein the signal decay of *cis*-pinonic acid was measured with the same instrument, before and after being modified to the EVR configuration. All other experimental parameters were kept identical. The signal decay time dropped from $\tau_{1/e}$ = 181 s to $\tau_{1/e}$ = 4 s. In the revised manuscript, we are presenting these data as the first results in the Results section, clearly demonstrating and highlighting the benefit of surface passivation.

*Lines 55-58 at the end of the introduction do not even mention the temperature issue, suggesting that the material changes are the main topic of this manuscript. This requires verification.*

We refer to our comment above, but agree that the effect of temperature should be mentioned in the Introduction. The revised paragraph reads as follows:" Herein we will demonstrate how the use of heated inlet capillaries made of passivated stainless steel (SS) and of a heated drift tube with passivated metal surfaces significantly improves the time response performance of PTR-MS analyzers. We will show that the heated and passivated instrument responds fast to low-volatility analytes, both for gas-phase and particle-phase measurements."

*A related point is the lack of any schematic diagram of the EVR in the manuscript. It may be understandable if the design itself did not change, but rather only ma-*

*terials were exchanged, yet it would still be beneficial for a reader to see a figure showing these changes. As it is, the only reference on PTR in section 2.1 is the Yuan et al (2017) review, which itself doesn't have a schematic of the exact system used in this work. This makes it very laborious for a reader to understand the changes, and consequently to properly assess the novelty of the changes and the manuscript itself.*

This is a valuable suggestion and we have included a figure (new Fig. 1) showing the drift tube and inlet system with all passivated surfaces in the Experimental section.

**Specific comments:**

*1. Lines 28-29: Since these measurements were done using NH4+ adducts, I don't think it should any longer be called "PTR-MS".*

PTR-MS instruments can nowadays be operated with different chemical ionization reagent ions ($H_3O^+$, $NO^+$, $O_2^+$, $NH_4^+$, others), and most of them do not react via proton transfer reactions (PTR). It is debatable if it makes sense to introduce additional acronyms for different operation modes of the same instrument. In the past, the acronym "eTR-MS" has been proposed for the $O_2^+$ mode but it was never taken up by the community. Our operation mode should probably be called "AAF-MS" (AAF: Ammonium Adduct Formation). We think that introducing new acronyms for different operation modes of the same instrument would just create confusion. Note that we are referring to the PTR-MS instrument and not to PTR-MS as a chemical ionization method.

*2. Lines 69-70. Since T_drift is used later as a parameter, it is important for a reader to have a clear picture of how the drift tube looks. Also here a schematic would be useful.*

As stated above, we have included a figure showing the drift tube and inlet system in the Experimental section.

*3. Lines 50-51: "H3O+ ion chemistry thus detects a wider spectrum of ana-lytes than any other chemical ionization method for atmospheric organic car-bon." This is a strong statement and would need a citation. Instruments like the NH4+-Cl3-TOF or C3H7NH3+-APi-TOF (Berndt et al., 2018, Angew. Chem. Int. Ed. 2018, 57, 3820 –3824) seem to detect almost all organic compounds (including radicals) except hydrocarbons. Can the authors show references where H3O+ ion chemistry would have detected a broader spectrum than that?*

Well, basic ion chemistry tells us that $H_3O^+$ ions react with a broader spectrum of organic species than $NH_4^+$ or $C_3H_7NH_3^+$ ions. The scope of this paper is, however, not to discuss the pros and cons of different reagent ions. Since Reviewer 2 also did not like our comparative statement, we have removed it.

*4. Lines 73-75. These two sentences need to be reformulated. "was measured as the time that evolved until" is hard to understand.*

We changed our wording to: "$\tau_{1/e}$ is the time it took the analyte signal to decay to 1/e (36.8%) of its initial value. $\tau_{90}$ is the time it took the analyte signal to decay to 10% of its initial value."

*5. Fig. S1. The part inside the red dashed box is presumably the inlet and drift tube of the PTR? This needs to be clarified, as it is not easy to read for someone not highly experienced with the system.*

Fig. S1 has been revised to better show the experimental set-up. Additional details are shown in the newly included Fig.1.

*6. Lines 108-109: This leaves the reader with the question "why?".*

It was more difficult to generate low volume mixing rations with these substances. This is now explained in the text.

*7. Lines 133-134: This also leaves the reader with the question "why?".*

The solid *cis*-pinonic acid sample was heated to 70 °C for generating a measurable concentration in the gas phase. Using lower instrument temperatures would have resulted in condensation on the walls. This is now explained in the text.

**8. Section 3.2: The discussion is about T_drift, and the text suggests that it is only the drift tube temperature that is changed. But Fig. S1 suggests that the entire inlet is one temperature-controlled entity. Please clarify. This again would be easier to understand if there was a proper schematic included.**

Fig. 1, which has been newly included in the Experimental section, should make it clear what parts are heated in the temperature-controlled instrument enclosure.

**9. Lines 138-139: This seems consistent, but Fig. S2 shows that the dependence of tau on C0 is very weak, with compounds of the same C0 easily having an order of magnitude or larger differences in tau. Were these two compounds selected to be shown because they happened to match?**

Both our data and literature data indicate that for long-chain ketones and carboxylic acids $\tau_{1/e}$ exhibits the expected increase with decreasing log $C^0$, while for saccharides and substituted phenols other factors seem to play a role. We thus show data for one long-chain ketone and one carboxylic acid. This is explained in the text:" For long-chain ketones (Pagonis et al., 2017; Krechmer et al., 2018) and carboxylic acids (Fig. S2), $\tau_{1/e}$ exhibits the expected increase with decreasing log $C^0$. Since log $C^0$ depends upon temperature, changes in T$_{drift}$ should lead to predictable changes in $\tau_{1/e}$. We thus measured $\tau_{1/e}$ for 2-tridecanone$_{(g)}$ and *cis*-pinonic acid$_{(g)}$ at variable T$_{drift}$ (Fig. 5)."

**10. Line 144: Why are you now shifting from tau_1/e to tau_90?**

Some of the literature data were only reported as $\tau_{90}$.

**11. Line 145: It is unclear which data from this study is used in Fig. 4. This should be made clear, so that a reader would be able to compare responses**

*compound by compound. In addition, there is a nice monotonic trend of tau vs C0 in Fig. 4, which is hard to understand given the huge spread of points in Fig. S2. One is left wondering how the authors selected the 5 data points shown in Fig. 4 for the EVR.*

The compound names have been included on the upper x-axis of the figure. Comparing the time response of different instruments to different compounds is obviously difficult, and one needs to be careful not to compare apples with pears. Our data suggest that at least for carbonyls and carboxylic acids the observed trends in $\tau_{1/e}$ can be explained by differences in $C_0$, which is why we have only included such compounds in the figure. We explain this in the revised text:" The upper horizontal axis lists the compound names; the lower horizontal axis shows in which log $C^0$ range molecules are classified as volatile organic compounds (VOCs), intermediate volatility organic compounds (IVOCs) and semi-volatile organic compounds (SVOCs). Since the instruments did not measure the same SVOCs, we only use our carboxylic acid data for the log $C^0$-based comparison."

*12. Line 153: C13 ketones are referenced, but I do not know where I should be looking to find this data.*

The compound names have been included on the upper x-axis of the figure.

*13. Fig. 4: It took me a while to realize that the unit of tau changes between Fig. 2 and Fig. 4, from sec to min. Why not keep them the same? Now both the time unit and the decay reference (1/e vs 90) change, and it makes things much harder to follow. I suggest to make all of these the same, as it would make the reading much smoother and avoid confusion.*

All times are now reported in seconds.

*14. Fig. 4: If I understand the plot correctly, the non-EVR PTR-TOF from this study seems to work much better than the EVR. All measured response times*

*are on the order of 0.01min, while in the EVR all points are at 0.1 min or higher.*
*Does this mean that the EVR setup has actually made the response times worse*
*compared to the original design (as long as the inlet is heated)?*

Unfortunately, the reviewer has misinterpreted the figure. log $C^0$ is a measure of
volatility and the compounds studied include volatile organic compounds (VOCs),
intermediate-volatility organic compounds (IVOCs) and semi-volatile organic com-
pounds (SVOCs). We have added a classification bar on the x-axis for including this
information. As a general trend, $_{90}$ increases with decreasing volatility or decreasing
$C_0$. What the figure shows is the following:

1. For VOCs and the more volatile IVOCs (log $C^0 \geq 5$), $\tau_{90}$ is close to the volumet-
   ric exchange time of the drift tube for both the EVR and the conventional PTR-
   ToF-MS instrument, as long as the drift tube is heated to 120 °C. Conventional
   unheated PTR-MS instruments have a much slower time response.

2. For the less volatile IVOCs (log $C^0 \leq 3$) and SVOCs, surface passivation reduces
   $\tau_{90}$ by two orders of magnitude (*cis*-pinonic acid), even if the drift tube is heated to
   120 °C in both the conventional and the EVR PTR-ToF-Ms analyzer. The heated
   EVR PTR-ToF-MS responds as fast as state-of-the-art CIMS instruments.

We have adapted the text to convey the above information.

*15. Line 155: The fact that temperature explains the major part of the differences,*
*and this fact is only mentioned in one sentence in the main text, makes Fig.*
*4 very misleading. For example, one would read from Fig. 4 that the reason*
*the PTR-TOF in this study was 3-4 orders of magnitude better than the PTR-*
*qMS from Pagonis et al is related to the quad vs tof, since that is the only clear*
*difference. Not to mention the comparison to other instruments, which were not*
*run at elevated temperatures. Please put the operating temperatures into the*

[Figure]

*figure legend, since these values are the most critical parameter to understand the major differences in the figure.*

Temperature is certainly an important factor and we have included the operating temperatures in the figure legend. The key information is, however, that for the less volatile IVOCs (log $C^0 \leq 3$) and SVOCs an increase in the drift temperature to 120 °C alone does not do the job. It is the surface passivation that drops $\tau_{90}$ by two additional orders of magnitude.

*16. Lines 171-172: Again, the fast changes are attributed to the materials, and temperature is not mentioned at all. This needs to be clearly validated before making this claim.*

We have extensively addressed this issue in previous comments and in the revised manuscript.

*17. Conflict of interest statement: Ionicon analytik is said to be "commercializing (CHARON) PTR-MS", but they are also advertising directly the EVR setup presented here (https://www.ionicon.com/accessories/details/extended-volatility-range-evr). Why is this not mentioned here?*

We are now explicitly mentioning that IONICON Analytik commercializes PTR-MS, CHARON and EVR.

**Technical corrections:**

*1. Line 97 and line 107: Caption, not legend.*

This was corrected.

---

## Author Comment (AC2) · 15 Nov 2020

**Response to Reviewer #2:**

We thank the reviewer for carefully reading our manuscript and for providing highly valuable comments, which we have addressed in detail below.

**Major comments:**

*1. The purpose and the focus of the manuscript are a little unclear. Is it just to estimate the signal-decay times for a new instrument or to assess its performance more holistically? If so, the effects of interactions with inlet walls and humidity should be discussed in this paper.*

[Figure]

The main purpose of this manuscript is to establish the EVR PTR-MS instrument in the scientific literature. More than two dozens of EVR PTR-MS instruments are nowadays in use, and we think it is important that the users can refer to an instrument paper when presenting their own data. We want to present a new type of PTR-MS instrument and exemplify its improved performance, rather than systematically investigating material, temperature or humidity effects.

*2. The effect of the drift tube temperature is interesting and important for more comprehensive evaluation of the instrument measurement capability, but is weakened by the very small number of compounds used to derive the conclusion that 120oC is the optimal temperature at which the instrument should be operated. It would be of interest to conduct similar measurements with a larger set of compounds, especially with the ones that tend to thermally decompose at higher temperatures, such as hydroperoxides (i.e., cumene hydroperoxide and dicumylperoxide discussed later in the paper).*

While the suggested thermal decomposition study would certainly be interesting, we feel that such work would go beyond the scope of a first instrument paper (see reply to comment 1). Please also note that we are not claiming that 120 °C is the optimal operation temperature. In fact, the operation temperature must be adapted to the type of analytes that are to be measured. We are explicitly stating this in the revised manuscript.

**3. I do not fully understand how detection of particle-phase highly oxidized organic compounds produced via ozonolysis of limonene fits in this paper. The authors neither discuss the signal-decay times for these compounds nor try to estimate the respective wall losses. It has been shown several times that softer ionization techniques, such as NH+ 4 CIMS, can be used for detection of highly oxygenated compounds in the gas and particle phase (i.e., Hansel et al., 2018; Zaytsev et al., 2019). Hence, the authors should clarify why they present these data and how implementation of the new inlet improves detection and quantifi-**

**cation of these compounds**.

We agree that we did not properly convey the information we wanted to give. Limonene ozonolysis was just a way for generating highly oxidized compounds up to $O_8$, which were not among the pure substances (up to $O_6$) available for our study. We have taken up the suggestion by the reviewer and show the signal decay of an $O_8$-compound. This demonstrates that even for highly oxidized analytes (ELVOC) $\tau_{1/e}$ remains below 20 s.

**Specific comments:**

*1. Lines 49-54: From reading this paragraph one might get a false impression that PTR-MS has the best measurement capability among all CIMS instruments. While it is true that the ionization efficiency does not vary too much among oxidized and nonoxidized compounds, the overall measurement capability of PTR-MS instruments is significantly limited by ionic fragmentation and wall losses. Hence, the authors should edit this paragraph and make their description of PTR-MS more balanced.*

We have only stated the $H_3O^+$ ion chemistry detects a broader spectrum of organic analytes than other CI techniques, which is certainly true from a pure ion chemistry point of view. This does *per se* not imply that PTR-MS has a better measurement capability, because the latter depends on additional factors (*e.g.*, inlet losses, detection limit). Since Reviewer 1 also did not like our comparative statement, we have removed it.

*2. Lines 55-58: The effects of drift tube temperature are not mentioned here, however they seem to play a fairly important role as outlined in the Discussion section. The particle-phase experiments should also be mentioned at the end of Introduction.*

This has also been pointed out by Reviewer 1 and we mention the effect of temperature in the revised paragraph. We also mention the particle experiments. The revised

paragraph reads as follows:" Herein we will demonstrate how the use of heated inlet capillaries made of passivated stainless steel (SS) and of a heated drift tube with passivated metal surfaces significantly improves the time response performance of PTR-MS analyzers. We will show that the heated and passivated instrument responds fast to low-volatility analytes, both for gas-phase and particle-phase measurements."

*3. Section 2.1: This section is missing a schematic of the EVR PTR-MS instrument. I suggest moving Fig S1 to the main text and significantly expanding it to demonstrate what parts of the instrument were replaced or coated. As of now, these changes might not be obvious especially for a reader who is not fully familiar with IONICON PTR-MS instruments.*

The drift tube and capillary inlet system (including the passivated parts) are sketched in Figure 1 of the revised manuscript.

*4. Line 95: Why did the authors use the double exponential decay for fitting the signals? How did the authors calculate τ 1/e from fitted parameters b1 and b2? I believe this is not explicitly discussed in the paper.*

We did not calculate $\tau_{1/e}$ from the fit. $\tau_{1/e}$ is simply the point in time when the analyte signal had dropped to 36% of its initial value. This is explicitly stated in the manuscript (§2.2). The fit was just included for guiding the eye; the fit function was included upon request of the editor.

*5. Figure 2: What do different circles/data points for the same compound represent? The authors should clarify this and discuss why the difference between some data points is fairly large, for example for 2,6-dimethoxyphenol and diglycolic acid it can be up to a factor of 2.*

See figure legend. Also see figure caption: „The size of the dots indicates the initial steady-state mixing ratio (0.1–100 ppbv) used in the respective experiment."

*6. Lines 140-141: It would be beneficial if the authors could include high-*

*resolution mass-spectra in the Supplement to demonstrate that studied compounds did not thermally decompose. Many of observed compounds are known to undergo ionic fragmentation (e.g., C6H9O + 4 is an ionic fragment of levoglucosan as discussed later in the paper), so how do the authors know that there is no additional thermal decomposition resulting in formation of those fragments?*

The reviewer raises a good point; our statement was too general. What we can state is that we did not observe any decarboxylation products. This has been corrected in the revised manuscript:" Exposing the sample gas to heated surfaces in an analyzer, may thermally degrade some analytes. It is important to note that none of the acids studied in this work decarboxylated at $T_{drift}$ = 120°C. It may, however, be necessary to use a lower $T_{drift}$ when more thermally labile analytes are targeted."

**7. Section 3.3: The authors should provide a table in which they should list compounds that were used to compare performances of various instruments. What ketones, carboxylic acids and hydroxycarbonyls were used in this study?**

The compound names have been included in the figure.

*8. Figure 4: It seems to me that the authors did not measure response times for the same compounds using a conventional PTR-MS and a new EVR PTR-MS as yellow and dark red points are located far from each other (for yellow dots 5<logC0<7.3 while for dark red dots 0.5<logC0<5). I suggest that the authors include additional data points on this figure to demonstrate how the performance of the EVR PTR-MS instrument compares with a conventional IONICON PTR-MS for the same group of compounds.*

We agree that including more data points would be valuable, but in this case adding more data from the conventional IONICON PTR-MS would not give additional information. We show that for 2-tridecanone (log $C^0 \sim 5$) the response is almost identical to that of the EVR-type instrument. Since the instrumental response time is already very close to the volumetric exchange time of the drift tube, we refrained from studying shorter-chain and thus more volatile ketones. We also show that for *cis*-pinonic acid (log $C^0 \sim 3$) $\tau_{90}$ is close to 2000 s for the conventional PTR-ToF-MS instrument, meaning that the three SVOCs we studied with the EVR-type instrument are simply not measurable with a conventional analyzer. It would certainly be interesting to compare the performance in the $3 < \log C^0 < 5$ range, but there we do not have any data (except for glycolic acid) and it would become a disproportionate effort to perform additional measurements study with two instruments.

**9. Figure 4: I agree with the Referee 1 that operational temperatures should be clearly stated in the legend of this figure.**

We have included the operational temperatures in the legend.

*10. Figure 6: The authors state that mass concentrations of observed compounds were calculated under the assumption that all of these compounds were detected at the collisional rate. The authors should clarify how this collisional rate was calculated. In addition, they should explicitly mention it in the text as this is a fairly important assumption and can strongly affect the authors' conclusion about mass yields of observed compounds.*

This figure has been moved to the Supplement and we provide additional details in the figure caption.

*11. Conflict of interest: I agree with the Referee 1 that the authors should mention the fact that IONICON has been advertising the EVR PTR-MS setup for quite some time now.*

We are now explicitly stating that IONICON Analytik commercializes PTR-MS, CHARON and EVR.

**Technical corrections:**

*1. Line 307: remove "in an"*

Done.

---

## Author Response (AR2)

**Response to Reviewer 1:**

*In response nr 15, the authors state that they have included operating conditions in the figure legend (current Fig. 6), but the legend in the revised version is identical to the original. This should be amended, but otherwise I suggest to accept as is.*

The reviewer has probably confounded the figure legend with the figure caption. The revised legend does indeed show the operating temperatures and is not identical to the original. In the revised version, we have included an additional note in the figure caption: "Note that the figure legend shows the different operating temperatures of the instruments used (RT: room temperature).

**Response to Reviewer 2:**

*I still have one question about Figure 2 (Figure 4 in the revised manuscript). What do different circles/data points for the same compound represent? I don't think this is mentioned in the caption or the legend. In my opinion, it would be beneficial to discuss why the difference between some data points is fairly large, for example for 2,6-dimethoxyphenol and diglycolic acid it can be up to a factor of 2.*

We do explain this in the figure caption ("The size of the dots indicates the initial steady-state mixing ratio (0.1–100 ppbv) used in the respective experiment.") and in the text ("We typically measured $\tau_{1/e}$ at three different mixing ratios for each compound.").

For further clarification, we provide a more detailed explanation in the text. "We typically measured $\tau_{1/e}$ at three different mixing ratios for each compound." → "We typically measured $\tau_{1/e}$ at three different mixing ratios for each compound. Three data points are thus typically shown for each analyte."

The reviewer has probably interpreted the three points per analyte as replicates. This is not the case and since the time response is concentration-dependent, we do expect significant variations when experiments are carried out at different analyte concentrations. The fact that we observed relatively large variations even at similar concentrations (*e.g.* for diglycolic acid as pointed out by the reviewer, but also for other low-volatility species) is most likely due to an additional passivation effect. We have added the following statement to the text: "Especially for low-volatility analytes, repeated sampling may passivate remaining active sites, which in turn improves the time response. This effect probably explains the relatively large variations in $\tau_{1/e}$ observed for the slow-responding analytes, even if all three experiments were carried out at similar mixing ratios."